# Graphene Scaffolds: A Striking Approach to Combat Dermatophytosis

**DOI:** 10.3390/nano13162305

**Published:** 2023-08-10

**Authors:** Shashi Kiran Misra, Himanshu Pandey, Sandip Patil, Tarun Virmani, Reshu Virmani, Girish Kumar, Abdulsalam Alhalmi, Omar M. Noman, Saad S. Alshahrani, Ramzi A. Mothana

**Affiliations:** 1School of Pharmaceutical Sciences, Department of pharmacy, Chhatrapati Shahu Ji Maharaj University, Kanpur 208024, India; 2Center for Teacher Education, Central Institute of Higher Tibetan Studies, Sarnath, Varanasi 221007, India; himanshu.nac@gmail.com; 3E-Spin NanoTech Private Ltd., SIDBI Innovation & Incubation Center, Indian Institute of Technology, Kanpur 208016, India; patilsandip13@gmail.com; 4School of Pharmaceutical Sciences, Modern Vidya Niketan University, Palwal 121105, India; reshu.virmani@mvn.edu.in (R.V.); girish.kumar@mvn.edu.in (G.K.); 5Department of Pharmaceutics, School of Pharmaceutical Education and Research, Jamia Hamdard, New Delhi 110062, India; 6Department of Pharmaceutical Biology, Institute of Pharmacy, University of Greifswald, 17489 Greifswald, Germany; omar.noman@stud.uni-greifswald.de; 7Department of Pharmacognosy, College of Pharmacy, King Saud University, Riyadh 11451, Saudi Arabia; ssalshahrani@ksu.edu.sa (S.S.A.); rmothana@ksu.edu.sa (R.A.M.)

**Keywords:** graphene nanoplatelets, electrospunned, dermatophytes, eudragit, dressing material

## Abstract

Exclusive physicochemical and biological properties of carbon allotrope graphene have attracted the peer attention of researchers for the synthesis and development of newer topical remedies including films, scaffolds, microspheres, and hydrogels. Here, graphene nanoplatelets (GN) were embedded into a different ratio of polymeric ERL100/ERS100 solution and fabricated in the form of a scaffold through the electrospinning process. FTIR spectra displayed characteristic similar peaks present both in GN and GN-loaded scaffold owing to the compatibility of GN and polymeric mixture. XRD curve revealed a distinct GN peak at nearly 26° whereas from DSC/TGA thermal stability was observed between polymers and graphene nanoplatelets. FESEM images showed ultrathin architecture of GN-loaded scaffold in a range of 280 ± 90 nm. The fabricated scaffold exhibited hydrophilicity (contact angle 48.8 ± 2.8°) and desirable swelling index (646% in skin pH media) which were desired criteria for the scaffold for topical application. In vitro, antifungal activity was conducted through the broth microdilution method against different virulent dermatophytes i.e., *Microsporum gypseum*, *M. canis*, *M. fulvum*, and *Trychophyton rubrum*. For in vivo evaluation, *T. rubrum* inoculum was applied on the dorsal surface of each group of Swiss albino mice, and the degree and intensity of mycelial growth or erythema on skin surfaces was visually investigated. The study depicted complete signs of cure after 14 days of application of G3-loaded scaffold on the infected dorsal site. Hence graphene-loaded scaffold represented a possible alternative for the treatment of topical fungal infections caused by dermatophytes.

## 1. Introduction

Carbon derivative graphene nanoplatelet (GN) is acquired from liquid phase exfoliation of graphite [1]. It has been offered as an ideal building block material for the scheming of newer therapeutic approaches owing to its inherent multiple contacts-based antimicrobial activities from the last decade [2,3,4,5]. Numerous investigations are being accomplished using sharp-edged GN for its potential to promote cell lysis that mediates antimicrobial activity. Its two-dimensional planer structure, narrow thickness (0.35 nm), small size (5 nm), large surface area (2630 m^2^/g), mechanical flexibility, and thermal conductivity with an appreciable young’s module (1TPa) endorse striking dynamics to develop and design diverse topical drug delivery systems for the management and alleviation of skin bugs/ailments caused by microorganisms [6,7,8,9,10,11]. Former studies reveal variable pathways of GN i.e., protein dysfunction, cell membrane damage, oxidative stress, plasma membrane perturbation, and transcriptional arrest to eliminate microorganisms [12,13,14].

Eukaryotic dermatophytes are physiologically dissimilar from prokaryotic bacteria and hence require a robust, potential, and controlled drug delivery scheme to provide instant and consistent bioactivity at the site of infection [15]. Dermatophytosis triggered by fungi that assault keratin tissues of the skin, hair, nails, feathers, and hooves of living beings, occurs/spreads in subtropical and humid regions [16]. The Tinea (group of infections) caused by filamentous [17], keratinophilic [18], and transmittable topical fungal bugs/dermatophytes, are generally ignored but can deteriorate the host immune system with time [19]. They consume keratinous substrates i.e., carbon, sulphur, and nitrogen through secretion of a proteolytic enzyme, and thus alter the topical infected site from an acidic environment to alkaline [20,21]. Reports say that approximately 25% of skin infections are associated with dermatophytes. Dermatophytosis is of high concern in health awareness and necessitates site-specific dermal remedies to have a significant cure rate and be protective as well [17,22].

Recently electrospunned Eudragit scaffolds have posed a new research interest by virtue of their structural organization [23,24,25,26], simulation of extracellular matrix [27], nice adherence at the site [28,29], superb absorption of release exudate [30], proper gaseous exchange during treatment, sufficient mechanical strength [31,32], and shielding from other infection [24]. Owing to being biocompatible, permeable, and non-sticky, polymethacrylate Eudragit polymers have been utilized to fabricate antimicrobial dressing materials [33,34,35]. We herein examined graphene nanoplatelets as a bioactive agent and electrospunned biocompatible Eudragit scaffolds intended for the alleviation of skin bug dermatophytes for the very first time.

## 2. Materials and Methods

### 2.1. Materials

Biocompatible polymers Eudragit RL100 and Eudragit RS 100 were received as gift samples from Evonik (Rohm Pharma, Darmstadt), Germany. Graphene nanoplatelet (GN) was purchased from Reinste Pvt. Ltd., Delhi, India. Contagious dermatophytes i.e., *Microsporum gypseum* (MTCC 2855), *Microsporum canis* (MTCC 2820), *Microsporum fulvum* (MTCC 2839), and *Trychophyton rubrum* (MTCC 7859) were purchased from the Microbial Type Culture Collection, Chandigarh, India. Roswell Park Memorial Institute Medium (RPMI 1640) and Morpholinepropanesulfonic acid (MOPS) were purchased from Sigma-Aldrich Chemicals Pvt. Ltd., New Delhi, India. Sabouraud Dextrose Agar (SDA) was purchased from HiMedia Laboratories Pvt. Ltd., Mumbai, India. N, N Dimethylacetamide, Methanol, and Dimethyl sulfoxide were acquired from Merck, India.

### 2.2. Fabrication of GN-Loaded Scaffolds

Two grades of biocompatible Eudragit polymers i.e., ERL 100/ERS 100 (total 15% *w*/*v*) at different ratios (1:3, 1:1, and 3:1) were selected for fabrication of scaffolds. 1% *w*/*v* GN was separately added into a different polymeric solution made in a solvent blend of Methanol and N, N Dimethylacetamide (60:40), subjected to magnetic stirring at room temperature for 8 h to prepare a well dispersed and non-aggregated mixture. The presence of hydroxyl and carboxylic groups in the chemical structure of GN must facilitate the formation of uniform dispersion in that system which was desirable for smooth conduct of electro spinning. Next, this GN-loaded dispersion was loaded into a 10 mL plastic syringe of 15 mm diameter equipped with a sharp needle (set flow rate 0.3 mL/h) and connected with a 15 kV voltage supply (Gamma High Voltage Research, Inc., Ormond Beach, FL, USA). As the voltage was increased, the repulsive electrical force pulled the developed pendent drop into a conical Taylor Cone that reached the bras collector (80 mm diameter) assembled in E-spin machine (E-Spin, Nanotech, IIT, Kanpur, India) within nanoseconds as regular fibers/or scaffolds [36,37]. Thereafter, collected scaffolds were kept in a desiccator till further investigation and characterization. Figure 1 portrays a schematic diagram of the fabrication of GN-loaded nanofibers through the electrospinning technique.

### 2.3. Characterizations

An attenuated total reflectance Fourier transform Infrared spectrometer (ATR-FTIR, Bruker Optick, Gmbh, Ettlingen and Leipzig, Germany) assembled with a DLaTNS detector with a germanium internal reflection element (IRE) crystal was employed at a range of 4000–500 cm^−1^ to recognize and characterize peaks of GN and its scaffolds by KBr disc method. Powdered X-Ray Diffraction studies (X’ Pert PRO, PAN analytical, Malvern, The Netherlands) of GN and scaffolds were executed with a 2θ range of 10–60° at 40 mV and 300 mA. The thermal behavior of fabricated scaffolds was investigated through a Thermogravimetric and Differential Thermal Analyzer (PerkinElmer, Pyris Diamond TG/DTA) assembled with resident Pyris Software (version 11). Approximately 5 mg of the sample was heated in a perforated and covered aluminum pan under the nitrogen purge from 50 °C to 600 °C with a heating rate of 10 °C/min.

### 2.4. Surface Morphology

Field emission scanning electron microscopy (FESEM, Quanta 200, Zeiss, Jena, Germany) was utilized to analyze the shape and size of all fabricated GN-loaded scaffolds (G1, G2, and G3). Before analysis, the scaffolds were gold sputter coated (up to 20 nm) under argon to reduce their electrical conductivity. Gold-coated samples were positioned in the microscopic cavity to which a high vacuum was applied, and their images were then captured at an excitation voltage of 15 kV. The dimension of approximately 30 fibers was analyzed and their mean average diameter was computed with image J software (https://imagej.net/ij/ accessed on 11 June 2023). The hydrophilic behavior of GN-loaded scaffolds was determined by measuring their water contact angle (WCA) through the sessile drop method (Rame Hart Inc., Succasunna, NJ, USA). The average WCA value/or extent of hydrophilicity was obtained by measuring the angle of the water droplets on the area of 5 cm^2^ scaffolds (each G1, G2, and G3) at five randomly distributed positions. Subsequently, swelling capacity was determined in triplicate manner by retaining each scaffold in 10 mL phosphate buffer media (pH 7.4) by gravimetric method for 24 h at 37 °C. Percentage swelling capacity was estimated by calculating the ratio of the weight of water adhered to the scaffolds to the weight of dried scaffolds.

### 2.5. In Vitro Antifungal Assay

Method of micro-dilution assay for anti-dermatophytic activity of fabricated G3 scaffold was executed following the Clinical and Laboratory Standards Institute guidelines (CLSI) against four contagious dermatophytes i.e., *M. gypseum* (MTCC 2855), *M. fulvum* (MTCC 2839), *M. canis* (MTCC 2820), and *T. rubrum* (MTCC 7859). One-week-old inoculated fungal strains were suspended in RPMI-1640 that was prior supplemented with L-glutamine and glucose (2%) to maintain pH 7 with Morpholinepropanesulfonic acid (MOPS, 0.165 mol/L). Prepared inoculums with 1–5 × 10^5^ CFU/mL were admixed in the flat-bottomed 96-well micro-titer plate with a serial dilution (0.01–1.25 mg/mL) of G3 nanofibers in dimethyl sulfoxide [37,38]. % Growth inhibition against time for each test pathogen was interpreted by computing the optical density of G3 scaffold employing a spectrophotometer (SPECTRAMAX plus 384, Molecular device, San Jose, CA, USA) at 530 nm for the period of 72 h at 37 °C without agitation.

### 2.6. In Vivo Study

The animal investigational protocol was permitted by the conducted ethical committee for the assessment of bioactivity (BU/Pharm/IAEC/15/01). The procedure operated in authorized accordance with the recommendations approved by the committee. Four-week-old male Swiss Albino mice of average weight 20–25 g were selected for an anti-dermatophyte study. Initially, the dorsal/topical surface of both groups was depilated with conventional hair removal cream. Dermatophyte *T. rubrum* was procured from MTCC, Chandigarh and selected for in vivo study. This infectious fungal inoculum was prepared from a 7-day-old culture and applied on the mouse dorsal surface after rinsing the surface with normal saline. The magnitude of mycelia growth of the fungal bug on mice’s skin was let to occur to form skin infection [39]. Topical treatment with G3 scaffold was started after infection of post inoculation and continued until complete recovery from the infection was achieved. Subsequently, the treatment and efficiency of the G3 scaffold were visually examined for two weeks.

### 2.7. Stability Study

Stability study for fabricated graphene embedded nanofibers was performed for three months and various parameters including nanofiber diameter, contact angle (a measure of hydrophilicity), percentage swelling index (absorption property), and in vitro antifungal responses against selected dermatophytes were evaluated.

## 3. Results

### 3.1. Fabrication of GN-Loaded Scaffolds

Carbon allotrope graphene GN was uniformly embedded in to blend of a total of 20% *w*/*v* ERL100: ERS100 compositions (1:3, 1:1, and 3:1), and their scaffolds (G1, G2, and G3) were efficiently fabricated and collected through electrospinning technique (E-Spin, Nanotech, IIT, Kanpur, India) applying stipulated organized procedural parameters. Selected solvent blend (methanol: N,N dimethylacetamide) allowed complete miscibility of ERL100/ERS100, which facilitated ease of Taylor cone creation and thus smooth conduct of electrospinning. Although the spinneret (tip of the syringe) was found to be clogged occasionally, this might be due to the high volatility of the solvent system and conductivity of GN.

### 3.2. Characterization

FTIR spectra of GN and its fabricated scaffolds displayed a characteristic peak at 3450 cm^−1^ due to the presence of O-H stretching vibrations and at 1650 cm^−1^ for C=O stretching vibrations (Figure 2a). At wavenumber 2950 cm^−1^, C–H stretching of the CH_3_ group was observed whereas C-OH stretching vibrations appeared at 1200 cm^−1^ which concludes the occurrence of traces of carboxyl groups in its chemical structure. However, stretching vibrations of C-O at 1060 cm^−1^ were observed in the spectra of the scaffold suggesting the presence of Eudragit polymers. Besides these, a strong C–H peak (out of plane) in GN was observed at 755 cm^−1^. A few distinctive peaks at 1240 cm^−1^ and 1750 cm^−1^ were visualized, relating C–O–C ester and C=O ester vibration groups present on ERL100/ERS100 polymers in the IR spectra of the scaffold.

Powder X-ray diffraction (PXRD) is an analytical tool primarily used for phase identification of crystalline material. In Figure 2b, an intensive sharp peak was obtained at 26° GN that suggested tidy arranged microcrystalline nature whereas PXRD of GN-loaded scaffolds could not change the highly crystalline nature of GN whereas the pattern exposed shifted peak at 18° may be due to its molecular dispersion between the blend of metastable polymethacrylate polymers i.e., ERL 100: ERS 100 in Figure 2b.

Thermogravimetric Analysis (TGA) was employed to measure the change in the percentage weight of GN and its scaffolds as a function of temperature in a controlled atmosphere. Figure 3a portrayed the TGA curve of GN that showed slight beginning of decomposition of graphene at 150 °C due to moisture loss and pyrolysis. Further, weight was consistently maintained up to 320 °C showing thermal stability of graphene nanoplatelets. A significant decrement of weight loss of GN was reported after 350 °C. GN-loaded nanofibers started decomposition at 260 °C and exhibited 9.2% weight loss owing to the polymeric degradation (Eudragit RL100/RS100).

Differential scanning calorimetry (DSC) was performed to analyze heat effects related to phase transitions and chemical reactions i.e., melting glass transitions, and recrystallization as a function of temperature. The addition of well-distributed GN in the scaffold not only acted as a strengthening agent which enhanced the firmness of the mesh/scaffold but also made a restriction of free molecular movement. Glass transition of GN was found to be quite higher (110 °C) than its fabricated scaffolds (95 °C). An endothermic peak at 343.5 °C of GN revealed its crystalline nature which was abolished in the scaffold must be due to interaction with Eudragit polymers better depicted in Figure 3b. However, thermal stability of GN loaded scaffold was found to be up to 450 °C after that molecular deformation was observed.

### 3.3. Surface Morphology

FESEM of electrospunned GN scaffolds i.e., G1, G2, and G3 revealed them to have smooth, elongated, nonwoven, and bead-free fibers. Although G1 exhibited irregularity with a little bit of broken architecture which must be due to its polymeric composition (high content of ERS 100), resultant diameter or average fiber size was scaled high as 403 ± 75 nm comparable to G2 (353.6 ± 85 nm) and G3 (280 ± 90 nm) fibers (Figure 4). Moreover, the existence of enormous, interconnected voids in G3 scaffolds would enable consistent release of bioactive GN from its architecture. Fabricated nano-scaled G3 scaffold certainly comprised great surface area that would be beneficial for the treatment and overwhelming of dermal bugs and site wounds. Additionally, Figure 4 of FESEM images exhibited no apparent sign of GN deposition at the surface suggesting the formation of uniform dispersion in polymeric solution and smooth conduct of electrospinning.

Water Contact Angle (WCA) is a consideration of wettability and hydrophilicity, specifying the manner a fluid dropped on a solid surface behaves and spreads out over there. Inset of Figure 4 of FESEM images scaffolds proposed the least contact angles of those scaffolds which have a higher content of ERL100 in their composition i.e., the higher extent of quaternary ammonium moiety. Fabricated GN embedded nanofibers G1, G2 and G3 possessed contact angles 67.5° ± 5.6°, 42.8° ± 4.5° and 48.2° ± 5.1° respectively. A perusal of the literature stated intrinsic water permeable feature of ERL100 favored wettability and thus hydrophilic scaffolds were obtained. Thus designed GN-embedded G2 and G3 scaffolds owned considerable wettability, and therefore would show efficiency to absorb released exudate during infection (dermatophytosis) which is a desirable criterion for dressing materials.

### 3.4. Swelling Index

Prodigious swelling behavior has been an obligatory norm for dressing materials for the sufficient absorption of ooze during topical pathogenic infections. Percentage swelling index values of graphene-loaded scaffolds were estimated in triplicate and comparative swelling profile was portrayed exhibiting standard deviation in Figure 5. G3 scaffolds exhibited the highest swelling index i.e., 646% compared to G1 (378%) and G2 (461%) scaffolds, which must be attributed to the presence of high content of polymer ERL 100. After 24 h, progressive deterioration on outcomes of the swelling index was achieved, which might be a loss in integrity of polymeric mesh/scaffolds (Figure 5).

### 3.5. Microdilution Assay

The G3 scaffold was selected for further experiments and analysis of the virtues of its fine architecture, narrow range dimension, high surface area, nice wettability, and impressive swelling index. Different transmissible dermatophytes (*M. gypseum* (MTCC 2855), *M. fulvum* (MTCC 2839), *M. canis* (MTCC 2820), and *T. rubrum* (MTCC 7859)) were selected for in vitro anti-dermatophytic study following microdilution assay. Figure 6 signifies percentage growth inhibition versus time for G3 scaffolds against pathogens for 96 h.

From the commencement of the study, scaffold G3 unveiled distinct vulnerability for *M. fulvum* (85.004%) which became consistent for the period of 96 h. However, after 72 h prominent growth inhibition/eradication of *T. rubrum* (89.703%) was also reported, that concluded elimination properties for both cited pathogens. Dermatophytes *M. gypseum* (65.888%) and *M. canis* (47.017%) were found to be less susceptible to the action of GN-embedded G3 scaffolds, suggesting their resistant nature and poor action of graphene. Hence G3 nanofibers were found to have antifungal activity against tenia pedis disease causative agent on humans as well as animals.

### 3.6. Stability Study

A detailed stability study by keeping nanofibers at 45 °C was performed for three months and various parameters including nanofiber diameter, contact angle (a measure of hydrophilicity), percentage swelling index (absorption property), and in vitro antifungal responses against selected dermatophytes were compiled. The outcomes suggested that there were no significant changes in the morphology and antifungal activities. Table 1 displays a comparative analysis of selected parameters in the duration of 0 months to 3 months. A slight increase in the diameter and contact angle of nanofibers was observed which might be due to storage and loss of solvent composition from the mesh (Figure 7). Similarly, decrement in the percentage swelling index may also be due to the dryness of the scaffold on storing at 45 °C. Antifungal activity against *T. rubrum* supported outcomes of previously performed microdilution assay before 3 months and displayed higher % growth inhibition compared to other strains of dermatophytes i.e., *M. fulvum*, *M. gypseum*, and *M. canis*.

### 3.7. In Vivo Study

Post inoculation with dermatophyte *T. rubrum* on Swiss albino mice exposed erythematous dorsal surface that got amplified on succeeding days (Figure 8). The figure showed comparative images of four groups i.e., Group I (control), Group II (blank, treated with only polymeric scaffolds), Group III (G3 treated) and Group IV (Marketed preparation) before and after treatment for two weeks. Determination of magnitude of infection was measured according to the number displayed in Table 2. The infected Groups I-IV scored 3 of magnitude of infection [40]. Group I did not receive any treatment and appearance of large red scaly and erythematous dorsal surfaces of mice exhibited the pathogenicity of *T. rubrum*, in Figure 8.

Group II mice received Eudragit RL/RS100 polymeric nanofibers and exhibited red and scaly patches that concluded ineffectiveness of polymers against this pathogen. By contrast, group III that were treated with G3 scaffolds exhibited a progressive cure rate of mice for seven days, and the appearance of new hair with healthy skin demonstrates effective treatment of G3 scaffolds. Similarly, marketed preparation was also found to be curative against *T. rubrum*. Finally, to test for complete eradication of *T. rubrum* from dorsal surface of mice, small quantities of skin were scraped from infected areas and cultured on SDA (Sabouraud Dextrose Agar) plate and incubated for 7 days to observe any kind of fungal colonies.

## 4. Discussion

Dermatophytes are the prime source of dermal infections leading to a wide range of diseases in different areas of the body. At the advanced and severe state of infection, conventional remedies prove rather helpless owing to their shortcomings related to frequent usage, biased drug releases, poor effectiveness and interaction with healthy tissues to the sufferers taking prolonged massive doses of antibiotics, corticosteroids, and cytotoxic agents. Nanoengineered GN platelets-embedded nonwoven fibers/scaffolds not only provided high surface area at the infection site but also exhibited localized effect without disturbing associated healthy tissues. The efficiency of nanocarrier GN for crossing the plasma membrane and promoting cellular uptake of the pathogen at infected site made an alternative to design dressing material for the eradication of dermatophytes [41,42]. Analytical tools such as FTIR and DSC/TGA demonstrated compatibility of GN with the blend of selected polymethacrylate polymers. Fabricated nanofibers exhibited nano-dimensional nonwoven bead-free interconnective nanofibers with optimum contact angle that are a desired benchmark for absorbing exudate. Amongst all, G3 nanofibers exhibited highest swelling index owing to its polymeric ERL100/ERS100 (3:1) composition and were subjected to microdilution assays against different strains of Tinea. In vitro study revealed maximum antidermatophytic action of G3 scaffolds for *T. rubrum* compared to other strains. In vivo studies confirmed efficacy of graphene-based nanofiber G3 on infected Swiss albino mice after 14 days.

## 5. Conclusions

This research aimed to explore bioactive GN as a recent strategy for the management of dermatophytosis. Effective cure rate from fabricated G3 scaffolds might be attributed to continuous diffusion of graphene nanoplatelets from scaffold at controlled rate that works as a depot at infected dorsal site and exhibits potential role of graphene nanoplatelets for regimen on dermatophytosis. A blend of biocompatible Eudragit polymers was selected for the fabrication of scaffolds owing to their superb permeability, nice adhesiveness, and controlled release behavior. The presence of quaternary ammonium groups in the molecular structure of Eudragit checks for secondary infection by their antimicrobial action. Nano-dimensional G3 scaffolds exhibited better hydrophilicity and swelling index, suggesting prospective potential to offer unblemished adhesion on the infected site as well as leading to augmented cure rate. Further studies are required to functionalize GN with antimicrobial therapeutics to provide a high payload, and steady and consistent action for an impressive regimen on tissue engineering, skin regeneration, and wound healing.

Findings also offered futuristic potential usage of polymethacrylate derivative Eudragit polymers for the development of biocompatible wound dressing materials (bandages/scaffolds) and by the attributes of their inherent quaternary ammonium compound composition.

## Figures and Tables

**Figure 1 nanomaterials-13-02305-f001:**
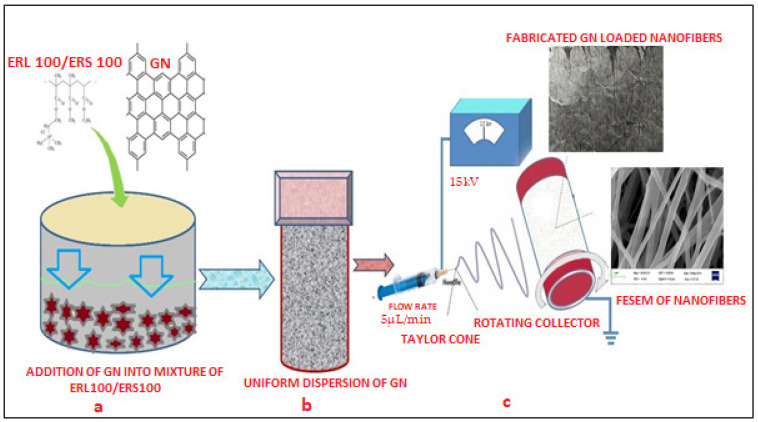
Schematic representation of the preparation of uniform polymeric dispersion GN (**a**,**b**) and fabrication of nanofibers through the electrospinning process (**c**).

**Figure 2 nanomaterials-13-02305-f002:**
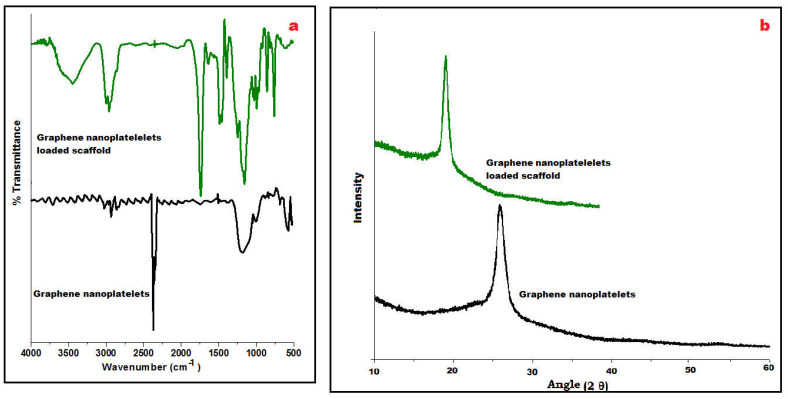
Compiled ATR-FTIR of graphene nanoplatelets and loaded scaffold revealing distinctive wavenumber (**a**), and PXRD of both displaying characteristic 2Ɵ angle (**b**).

**Figure 3 nanomaterials-13-02305-f003:**
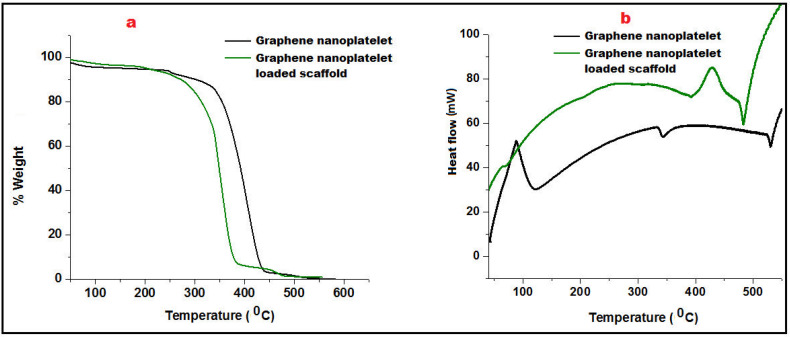
Comparative Thermo-gravimetric analysis (**a**) and Differential Scanning Calorimetry of Graphene nanoplatelet and graphene nanoplatelet loaded scaffold (**b**).

**Figure 4 nanomaterials-13-02305-f004:**
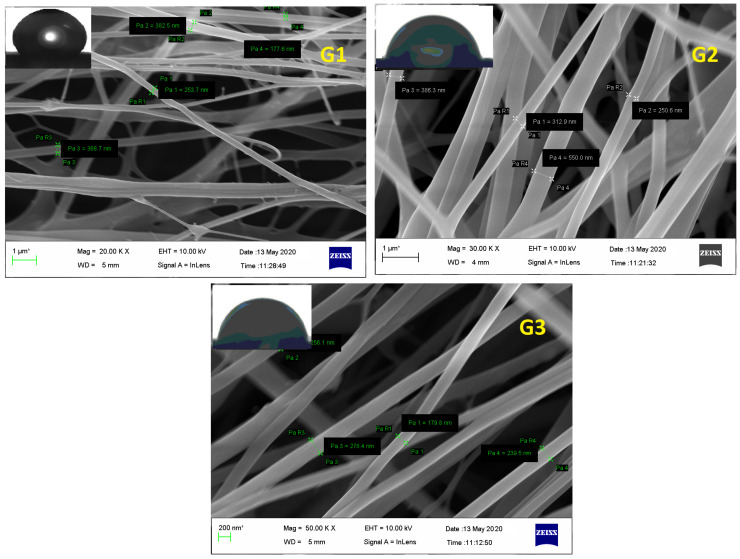
FESEM of GN-loaded scaffolds i.e., G1, G2, and G3 composed of biocompatible polymers ERL100: ERS 100 (1:3, 1:1, and 3:1 respectively).

**Figure 5 nanomaterials-13-02305-f005:**
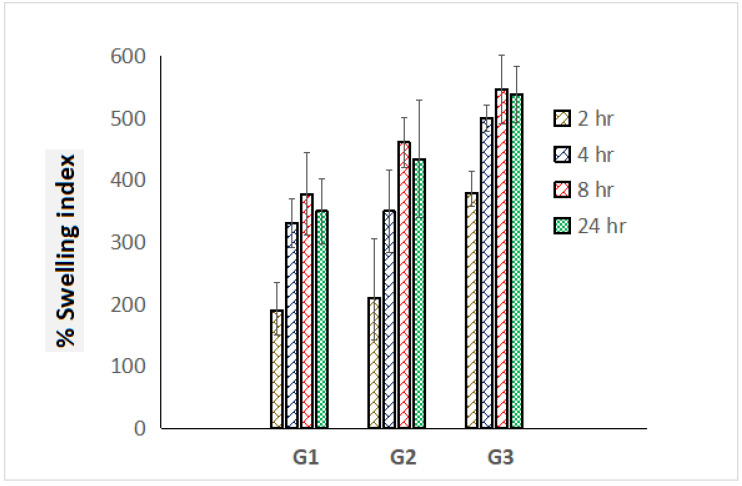
Comparative percentage swelling index of G1, G2, and G3 scaffolds for the time course of 24 h (measurements computed for three readings).

**Figure 6 nanomaterials-13-02305-f006:**
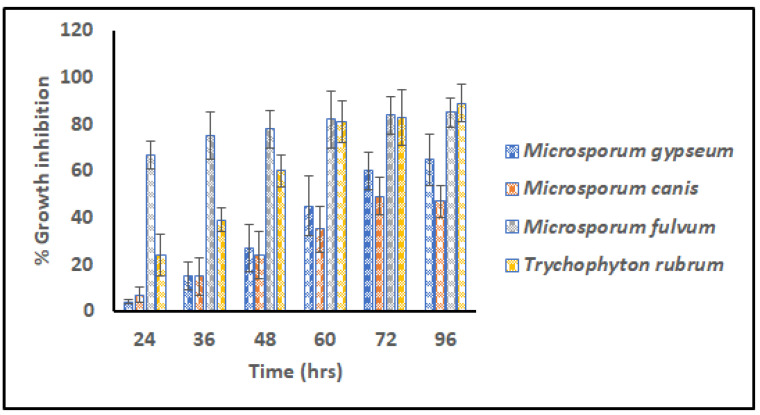
Relative Microdilution assay representing % growth inhibition of G3 scaffolds against contagious dermatophytes i.e., *M. gypseum, M. canis, M. fulvum,* and *T. rubrum* (Measurements computed for three values).

**Figure 7 nanomaterials-13-02305-f007:**
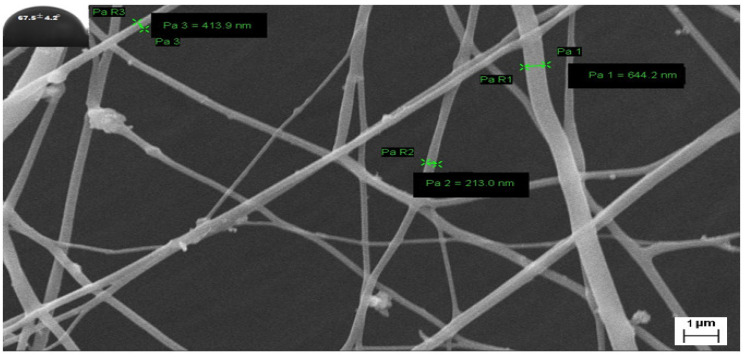
FESEM image of G3-loaded nanofibers and their contact angle after storing 3 months at 45 °C.

**Figure 8 nanomaterials-13-02305-f008:**
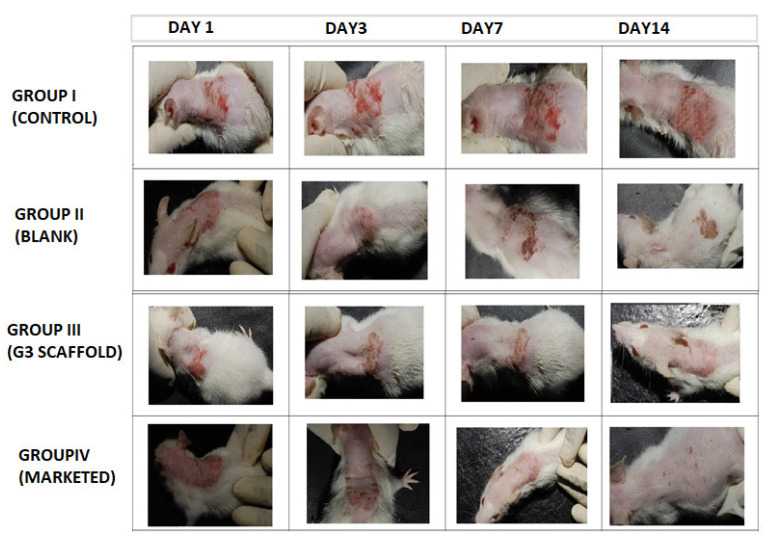
Comparative animal study elucidating severity of dermatophytosis on infected Swiss albino mice (Group I, Control), (Group II, Blank), (Group III, G3), (Group IV, Marketed) and successive post-treatment images after 14 days.

**Table 1 nanomaterials-13-02305-t001:** Compiled evaluated parameters for the stability study.

Parameters	Diameter of Nanofiber	Contact Angle	Percentage Swelling Index	In Vitro Antifungal Responses			
	G3 Nanofiber (nm)	G3 Nanofibers	%	*M. gypseum*	*M. fulvum*	*M. canis*	*T. rubrum*
0 month	280 ± 90 nm	48.8 ± 2.8°	646 ± 23%	65.888 ± 11.45%	85.004 ± 6.34%	47.017 ± 7.03%	89.703 ± 8.06%
3 months	423 ± 60 nm	67.5 ± 4.2°	426 ± 19%	60. 564 ± 8.88%	81.002 ± 12.34%	43.419 ± 7.67%	86.453 ± 9.54%

**Table 2 nanomaterials-13-02305-t002:** Determination of score according to magnitude of lesion on infected mice.

Score	Magnitude of Lesion
0	Absence of lesion
1	Appearance of erythema at infection site
2	Moderate erythema and spread
3	Erythema, abrasion and scaling
4	Severe erythema with lesion and scars

## Data Availability

Not applicable.

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
