# Peer review of "Graphene Scaffolds: A Striking Approach to Combat Dermatophytosis"

_nanomaterials, 2023, doi:10.3390/nano13162305_

Round 1
Reviewer 1 Report
In ‘Graphene Scaffolds: A Striking Approach to Combat Dermatophytosis’, Misra et al. prepared electrospun scaffolds of graphene nanoplatelets in ERL100/ERS 100 polymers and characterized them by FTIR, XRD, DSC, TGA, SEM, contact angle, swelling, and antifungal activity in vitro and in vivo. While these are reasonable experiments to perform, the reporting is very unclear and the study design lacks the expected replicates to confirm the results.
Specific comments:
- The whole text should be carefully edited for proper use of scientific terminology and phrasing.
- In the abstract it would be helpful to provide more detail about the in vivo model used.
- In line 43, carbon nanotubes and graphene nanoplatelets are different things.
- In line 47, it is not clear what “infrequent cell lysis-leading antimicrobial activity” means.
- Line 50: Young’s modulus is the correct term.
- Line 53: What is meant by “prose perusal”?
- Line 64: What does “With an alarming global burden (25%)” mean?
- Section 2.2: The concentration of polymer in the electrospinning solution is not specified.
- In section 2.5, the assay is not clearly described, especially lines 145-151. Likewise section 2.6, lines 158-162.
- The stability study in section 3.7 is not described in the methods.
- The interpretation of the TGA and DSC results in lines 197-217 is difficult to follow.
- Is the contact angle in Figure 4a correct?
- The swelling and microdilution experiments are lacking replicates. It is usually good experimental design to perform at least 3 independent replicates and report the average and standard deviation of the measurements. Also, is the antifungal experiment so sensitive to be able to report so many significant digits as in Table 1?
- The in vivo study needs some quantification of the extent of infection, as the results shown in the images are very qualitative. Also, only results with one animal per group are reported.
- The results are completely lacking a discussion of their relevance to what is reported with other materials in the literature.
Overall, the paper is written very informally.
Author Response
Reviewer1
Comment 1. The whole text should be carefully edited for proper use of scientific terminology and phrasing.
Answer 1: Dear reviewer, as per your suggestion the whole manuscript is revised to avoid typographical errors.
Comment 2- In the abstract it would be helpful to provide more detail about the in vivo model used.
Answer 2. The abstract is modified and in vivo results are more emphasised there.
Comment 3- In line 43, carbon nanotubes and graphene nanoplatelets are different things.
Answer 3. Yes reviewer, both are different now the sentence is corrected.
Comment 4- In line 47, it is not clear what “infrequent cell lysis-leading antimicrobial activity” means.
Answer4. The sentence is reframed now the meaning is clear.
Comment 5- Line 50: Young’s modulus is the correct term.
Answer 5. It is corrected there.
Comment 6- Line 53: What is meant by “prose perusal”?
Answer 6. Dear reviewer, the sentence is rectified with appropriate term in manuscript.
Comment 7- Line 64: What does “With an alarming global burden (25%)” mean?
Answer 7. The sentence is reframed here.
Comment 8- Section 2.2: The concentration of polymer in the electrospinning solution is not specified.
Answer 8. Polymer concentration was selected 15 %w/v and now it is mentioned in the section 2.2 of manuscript.
Comment 9- In section 2.5, the assay is not clearly described, especially lines 145-151. Likewise section 2.6, lines 158-162.
Answer 9. Both in vitro antifungal study and in vivo animal study are now explained in detailed way. Elaborate procedures are mentioned in manuscript also.
Comment 10- The stability study in section 3.7 is not described in the methods.
Answer 10. In material and method, section 2.7 Stability study is now added in manuscript.
Comment 11- The interpretation of the TGA and DSC results in lines 197-217 is difficult to follow.
Answer 11. Dear reviewer, both the studies i.e. DSC and TGA were carried out to explain nature of crystallinity and effect of temperature respectively on the graphene nanoplatelets embedded nanofibers. These studies defined thermal stability of graphene and its deviation after fabrication of polymeric scaffolds. Graphene is highly thermostable allotrope of carbon and shows excellent stability at high temperature.
Comment 12- Is the contact angle in Figure 4a correct?
Answer 12. Dear reviewer, now contact angle of image 4a is corrected.
Comment 13- The swelling and microdilution experiments are lacking replicates. It is usually good experimental design to perform at least 3 independent replicates and report the average and standard deviation of the measurements. Also, is the antifungal experiment so sensitive to be able to report so many significant digits as in Table 1?
Answer 13. Dear reviewer, as per your valuable suggestion, both images i.e. swelling index and microdilution assay have been corrected. Now the results are shown on the outcomes received from triplicate study. Moreover, table I comprised of comparative parameters utilized for evaluation of G3 nanofibers after 3 months. Hence the findings of antifungal are reported there.
Comment 14- The in vivo study needs some quantification of the extent of infection, as the results shown in the images are very qualitative. Also, only results with one animal per group are reported.
Answer 14. The image of in vivo study is now modified and exhibiting comparative treatment with control, blank, G3 scaffold and marketed preparation. The animal study was based on visual analysis.
Comment 15- The results are completely lacking a discussion of their relevance to what is reported with other materials in the literature.
Answer 15. The result and discussion part are modified and it highlights on relevance of study.

Reviewer 2 Report
1. The photos in the article need to be processed, many of them have been stretched.
2. The scale value of SEM photos cannot be seen clearly.
3. In figure 2, Why does the peak of XRD shift significantly?
4. The standard card peak for XRD should be provided.
5. What are the special uses of this one-dimensional structure?
need to be improved.
Author Response
Reviewer 2
Comment 1. The photos in the article need to be processed, many of them have been stretched.
Answer 1. Dear reviewer as per your suggestions the images are reanalysed and processed accordingly in the manuscript.
Comment 2. The scale value of SEM photos cannot be seen clearly.
Answer 2. Authors have modified FESEM image. Now clear scale can be seen.
Comment 3. In figure 2, why does the peak of XRD shift significantly?
Answer 3. Dear reviewer, in figure 2b, XRD peak of graphene embedded scaffolds shifts due to due to its molecular dispersion between the blend of metastable polymethacrylate polymers i.e., ERL 100: ERS 100. By contrast, an intensive sharp peak was displayed of graphene nanoplatelets might be due to microcrystalline nature of it.
Comment 4. The standard card peak for XRD should be provided.
Answer 4. Authors are thankful to the reviewer for their valuable inputs. For your kind reference, we are attaching the XRD of standard card.
Comment 5. What are the special uses of this one-dimensional structure?
Answer 5. One dimensional structure of nanofibers exhibits unique and exclusive shape-dependent attributes that enables control delivery of therapeutics and better attachment to the site specially when framed in a sheet. These structures are better alternatives for topical drug delivery systems compared to conventional cream, ointment and gels.
At last, all authors are paying their sincere thanks to the reviewers for his valuable suggestions and remarks. It surely upgraded the worth of this manuscript.

Reviewer 3 Report
The authors of this study have focused on synthesizing and characterizing graphene-loaded scaffolds. They investigated the compatibility between polymers and graphene nanoplatelets, assessed the physicochemical properties of the scaffold, and explored its potential for topical application. The findings suggest that graphene-loaded scaffolds could serve as a promising alternative for treating fungal infections caused by dermatophytes.
1. It is indeed an interesting study. However, there are some aspects that could be further improved and expanded upon. Firstly, it would be useful to present the drug release properties of the GN-loaded scaffold (G3). Understanding the release kinetics and profile of drugs from the scaffold is important in evaluating its potential as a drug delivery system.
2. I recommend conducting cell culture studies to evaluate the biocompatibility of the GN-G3 scaffolds. Assessing the response of relevant cell lines to the scaffold, such as skin cells or other appropriate cell types, would provide valuable insights into its compatibility with living tissue.
3. It would be beneficial to perform a toxicity and stability study to evaluate the safety of the GN-G3 scaffold. Investigating the potential adverse effects and assessing the scaffold's stability under different storage conditions would contribute to a comprehensive safety assessment.
4. The study lacks sufficient biological efficacy evaluation. Providing statistical significance for the in vitro antifungal assay and in vivo study is essential. This would help validate the effectiveness of the scaffold in inhibiting the growth of dermatophytes and provide a basis for further comparisons and conclusions.
5. It is recommended to present the results of the in vivo study as tables or figures. This would facilitate a clearer understanding of the experimental outcomes and enhance the visual presentation of the data.
Author Response
Reviewer 3
Comment 1. It is indeed an interesting study. However, there are some aspects that could be further improved and expanded upon. Firstly, it would be useful to present the drug release properties of the GN-loaded scaffold (G3). Understanding the release kinetics and profile of drugs from the scaffold is important in evaluating its potential as a drug delivery system.
Answer 1. Dear reviewer thanks for your kind suggestion. We are working on conjugation of graphene with different antifungal therapeutics and their comparative drug release profile to evaluate potential for anti-dermatophytic action. The current study focuses efficiency of graphene embedded scaffolds against devastating dermatophytes.
Comment 2. I recommend conducting cell culture studies to evaluate the biocompatibility of the GN-G3 scaffolds. Assessing the response of relevant cell lines to the scaffold, such as skin cells or other appropriate cell types, would provide valuable insights into its compatibility with living tissue.
Answer2. Dear reviewer, we welcome your suggestion and surely carry out cell culture study for termination of biocompatibility with skin cells.
Comment 3. It would be beneficial to perform a toxicity and stability study to evaluate the safety of the GN-G3 scaffold. Investigating the potential adverse effects and assessing the scaffold's stability under different storage conditions would contribute to a comprehensive safety assessment.
Answer 3. Dear reviewer, the present works contained stability study in various storage conditions. The outcomes collected from stability studies suggested no significant changes in the morphology and antifungal activities of fabricated graphene loaded scaffolds.
Comment 4. The study lacks sufficient biological efficacy evaluation. Providing statistical significance for the in vitro antifungal assay and in vivo study is essential. This would help validate the effectiveness of the scaffold in inhibiting the growth of dermatophytes and provide a basis for further comparisons and conclusions.
Answer 4. Dear reviewer, in vivo study was performed in four weeks old male Swiss Albino mice (average weight 20- 25 g) for an anti-dermatophyte study. A detailed procedure and outcomes are embedded in the manuscript. The results advocated in vitro results and displayed in vitro in vivo (IVIVC) correlation. G3 loaded nanofibers has exhibited effectiveness for inhibition of growth of infection and a progressive cure rate of mice for 14 days. Growth of new hair with healthy skin demonstrated better alternative for the treatment with graphene scaffolds.
Comment 5. It is recommended to present the results of the in vivo study as tables or figures. This would facilitate a clearer understanding of the experimental outcomes and enhance the visual presentation of the data.
Answer 5. Authors are thankful for the suggestion. A comparative treatment with control, blank, G3 scaffolds and marketed preparation is displayed in figure 8 in the manuscript. The image is clearly mentioning the effectiveness of applied G3 scaffolds on the infected site of mice.

Round 2
Reviewer 1 Report
The authors have addressed some comments from the reviewers and somewhat improved the study. I still have the following comments:
- The explanation of the TGA and DSC results is still unclear. As a specific example, "Figure 3a portrayed the TGA curve of GN that showed the primary decomposition of graphene at 150 °C followed by secondary prominent weight loss of up to 3.5%. The major weight loss at approximately 200 °C attributed to pyrolysis of the oxygen-containing functional groups and generation of Carbon- dioxide.As expected, the TGA of GN displayed steady thermal stability throughout the analysis of its chemical nature. By conrast, little weight loss might be at below100ËšC attributed to bound moisture, small traces 235 of organic solvents, and adherence of impurity if any. A great percentage weight loss (9.2 %) of GN scaffolds was started at 350ËšC, owing to the polymeric decomposition in the 237 scaffold which was found to completely disintegrate after a temperature of 450ËšC. " These statements are inconsistent. How is GN undergoing primary decomposition but also showing thermal stability?
- The water contact angle images should have an overlay of the measured angle, as the numbers do not seem to match well with the images.
- The number of replicates should be reported in the figure captions.
- The study is lacking statistical analysis of the data.
- It is still difficult to believe that the antifungal activity can be precisely measured to 0.00X %. Is it correct to be reporting the figures with this number of significant digits?
- The in vivo studies still appear to be lacking replicates and any sort of quantitative metrics.
- The study is still lacking discussion of the results in the context of the literature.
The writing still requires major improvements regarding English grammar.
Author Response
Reviewer comments
Comment 1- The explanation of the TGA and DSC results is still unclear. As a specific example, "Figure 3a portrayed the TGA curve of GN that showed the primary decomposition of graphene at 150 °C followed by secondary prominent weight loss of up to 3.5%. The major weight loss at approximately 200 °C attributed to pyrolysis of the oxygen-containing functional groups and generation of Carbon- dioxide.As expected, the TGA of GN displayed steady thermal stability throughout the analysis of its chemical nature. By conrast, little weight loss might be at below100ËšC attributed to bound moisture, small traces 235 of organic solvents, and adherence of impurity if any. A great percentage weight loss (9.2 %) of GN scaffolds was started at 350ËšC, owing to the polymeric decomposition in the 237 scaffold which was found to completely disintegrate after a temperature of 450ËšC. " These statements are inconsistent. How is GN undergoing primary decomposition but also showing thermal stability?
Answer 1: Authors are thankful to the reviewers for their valuable suggestions. As per your suggestions, the whole paragraph is altered and corrected.
Comment 2- The water contact angle images should have an overlay of the measured angle, as the numbers do not seem to match well with the images.
Answer 2- We are thankful to the reviewers for their insightful inputs. The contact angle was measured through Ram Hart Goniometer, and was received as it is. The angles are now removed from the image and discussed in the text.
Comment 3- The number of replicates should be reported in the figure captions.
Answer 3- As per your valuable suggestions, it is included in the appropriate figures.
Comment 4- The study is lacking statistical analysis of the data.
Answer 4- Authors are very thankful to the reviewers for their suggestions. As per your suggestions, statistical estimations are incorporated where ever applicable in manuscript.
Comment 5 - It is still difficult to believe that the antifungal activity can be precisely measured to 0.00X %. Is it correct to be reporting the figures with this number of significant digits?
Answer 5- The insightful comments of the reviewer will definitely improve the quality of manuscript. Dear reviewer, the antifungal study was carried out via SPECTRAMAX plus 384, Molecular device. It is highly sensitive and measures concentration of test sample up to three digits after decimal. Here attached optical density of 96 well plate and percentage growth inhibition of a test sample that showed concentration up to 0.00x%.
Comment 6- The in vivo studies still appear to be lacking replicates and any sort of quantitative metrics.
Answer 6- Authors are very thankful to the reviewers for their insightful comments. The animal study was carried out by taking six mice in a group. The related animal ethical committee approval letter has been shared to the editorial office. As per this study concern, one compiled image is prepared that showed effect of treatment on one mice of respective groups.
For measurement of effectiveness of applied test samples, treatment scores were visually observed. Here we attach two tables that were framed for defining magnitude of lesion and treatment score respectively. The visual examination method was adopted to determine the percentage of antifungal activity of applied test samples on the infection sites of each group. Supporting reference is included in manuscript.
Table 1. Determination of score according to magnitude of lesion on infected mice
|
Score |
Magnitude of lesion |
|
0 |
Absence of lesion |
|
1 |
Appearance of erythema at infection site |
|
2 |
Moderate erythema and spread |
|
3 |
Erythema, abrasion and scaling |
|
4 |
Severely erythema with lesion and scars |
Table 2. Estimation of treatment score based on cure rate
|
TREATMENT SCORE |
CURE RATE |
|
0 |
Uncured |
|
1 |
25% cure |
|
2 |
50 % cure |
|
3 |
75 % cure |
|
4 |
100 % cure |
Comment 7 - The study is still lacking discussion of the results in the context of the literature.
Answer7 –As per your recommendation, discussion part is added in the manuscript.
All authors are highly thankful for your kind suggestions that definitely would improve the quality of this research paper.

Author Response
Dear reviewer
We are thankful to you for accepting our changes in the manuscript.
Round 3
Reviewer 1 Report
While the authors have improved the manuscript, there are still some concerns:
- The interpretation of the TGA data still does not make sense. One problem might be that the y-axis in Figure 3 is labeled as "weight loss (%)" when it is most likely showing "weight (%)". The curves are almost flat at 150C, so it is not clear how this can be interpreted as showing decomposition, for example.
- The discussion is missing references, for example, for statements like: "The efficiency of nanocarrier GN for crossing the plasma membrane and promoting cellular uptake of the pathogen at infected site made an alternative to design dressing material for the eradication of dermatophytes." The present study did not measure these things.
- The conclusion that the "effective cure rate from fabricated G3 scaffolds might be attributed due to continuous diffusion of graphene nanoplatelets from scaffold at controlled rate that worked as depot at infected dorsal site and exhibited potential role of graphene nanoplatelets for regiment on dermatophytosis." is not supported by the results. No measurement of graphene nanoplatelet release was performed.
The manuscript still requires revision with respect to English grammar and word choice.
Author Response
Reviewer comments
Comment 1- The interpretation of the TGA data still does not make sense. One problem might be that the y-axis in Figure 3 is labeled as "weight loss (%)" when it is most likely showing "weight (%)". The curves are almost flat at 150C, so it is not clear how this can be interpreted as showing decomposition, for example.
Answer 1- Authors are thankful to the reviewer for his valuable observation. As per your suggestion, we have corrected the Y-axis of figure 3a as % Weight. The interpretation of TGA is also rectified according to your suggestions.
Comment 2- The discussion is missing references, for example, for statements like: "The efficiency of nanocarrier GN for crossing the plasma membrane and promoting cellular uptake of the pathogen at infected site made an alternative to design dressing material for the eradication of dermatophytes." The present study did not measure these things.
Answer2- We are thankful to the reviewer for his insightful suggestions. As per your suggestion, appropriate references are embedded in this section. Further, these statements suggested possible mechanism of graphene nanoplatelets while interacting with pathogens (here dermatophytes).
Comment 3 - The conclusion that the "effective cure rate from fabricated G3 scaffolds might be attributed due to continuous diffusion of graphene nanoplatelets from scaffold at controlled rate that worked as depot at infected dorsal site and exhibited potential role of graphene nanoplatelets for regiment on dermatophytosis." is not supported by the results. No measurement of graphene nanoplatelet release was performed.
Answer3- Authors are thankful to the reviewer for his valuable suggestion. Fabricated nanofibers have been loaded with graphene nanoplatelets (GN) and their exclusive mesh like architecture and hydrophilicity would tend to diffuse out GN on coming contact with released exudate (alkaline pH 7.2-8.9) at the site of infection. Polymethacrylate polymers showed swelling at this pH without degradation and suggested the controlled release behaviour of embedded GN from the nonwoven mesh. Moreover, as no therapeutic agent was added with GN so only in vitro antifungal activity (microdilution assay) was performed that is confirming potential of GN against dermatophytes.
Comment 4-The manuscript still requires revision with respect to English grammar and word choice.
Answer 4- As per your suggestions, the manuscript was again revised and found some typographical errors that have been rectified.
We all authors are very thankful to the reviewer for his/her keen observation for this manuscript. We have learnt a lot and tried to rectify related issues.
